# Genomic and Phylogenetic Characterisation of SARS-CoV-2 Genomes Isolated in Patients from Lambayeque Region, Peru

**DOI:** 10.3390/tropicalmed9020046

**Published:** 2024-02-11

**Authors:** Sergio Luis Aguilar-Martinez, Gustavo Adolfo Sandoval-Peña, José Arturo Molina-Mora, Pablo Tsukayama-Cisneros, Cristian Díaz-Vélez, Franklin Rómulo Aguilar-Gamboa, D. Katterine Bonilla-Aldana, Alfonso J. Rodriguez-Morales

**Affiliations:** 1Grupo de Investigación en Bioinformática y Biología Estructural, Facultad de Ciencias Biológicas, Universidad Nacional Mayor de San Marcos, Lima 15081, Peru; sergioluis.aguilar@unmsm.edu.pe (S.L.A.-M.); gsandovalp@unmsm.edu.pe (G.A.S.-P.); 2Centro de Investigación en Enfermedades Tropicales, Facultad de Microbiología, Universidad de Costa Rica, San Jose 11501-2060, Costa Rica; jose.molinamora@ucr.ac.cr; 3Facultad de Ciencias e Ingeniería, Universidad Peruana Cayetano Heredia, Lima 15102, Peru; pablo.tsukayama@upch.pe; 4Facultad de Medicina, Universidad Privada Antenor Orrego, Trujillo 13008, Peru; cdiazv3@upao.edu.pe; 5Grupo de Investigación en Inmunología y Virología del Norte, Laboratorio de Inmunología y Virología, Hospital Regional Lambayeque, Lambayeque 14011, Peru; faguilar@hrlamb.gob.pe; 6Research Unit, Universidad Continental, Huancayo 12001, Peru; 7Faculty of Health Sciences, Universidad Científica del Sur, Lima 15307, Peru; arodriguezmo@cientifica.edu.pe; 8Gilbert and Rose-Marie Chagoury School of Medicine, Lebanese American University, Beirut P.O. Box 13-5053, Lebanon

**Keywords:** SARS-CoV-2, genome, sequencing, phylogenetic analysis, Peru

## Abstract

Objective: this study aims to identify and characterise genomic and phylogenetically isolated SARS-CoV-2 viral isolates in patients from Lambayeque, Peru. Methods: Nasopharyngeal swabs were taken from patients from the Almanzor Aguinaga Asenjo Hospital, Chiclayo, Lambayeque, Peru, which had been considered mild, moderate, and severe cases of COVID-19. Patients had to have tested positive for COVID-19, using a positive RT-PCR for SARS-CoV-2. Subsequently, the SARS-CoV-2 complete viral genome sequencing was carried out using Illumina MiSeq^®^. The sequences obtained from the sequence were analysed in Nextclade V1.10.0 to assign the corresponding clades, identify mutations in the SARS-CoV-2 genes and perform quality control of the sequences obtained. All sequences were aligned using MAFFT v7.471. The SARS-CoV-2 isolate Wuhan NC 045512.2 was used as a reference sequence to analyse mutations at the amino acid level. The construction of the phylogenetic tree model was achieved with IQ-TREE v1.6.12. Results: It was determined that during the period from December 2020 to January 2021, the lineages s C.14, C.33, B.1.1.485, B.1.1, B.1.1.1, and B.1.111 circulated, with lineage C.14 being the most predominant at 76.7% (n = 23/30). These lineages were classified in clade 20D mainly and also within clades 20B and 20A. On the contrary, the variants found in the second batch of samples of the period from September to October 2021 were Delta (72.7%), Gamma (13.6%), Mu (4.6%), and Lambda (9.1%), distributed between clades 20J, 21G, 21H, 21J, and 21I. Conclusions: This study reveals updated information on the viral genomics of SARS-CoV-2 in the Lambayeque region, Peru, which is crucial to understanding the origins and dispersion of the virus and provides information on viral pathogenicity, transmission and epidemiology.

## 1. Introduction

COVID-19 is a respiratory disease caused by the SARS-CoV-2 virus (severe acute respiratory syndrome coronavirus 2), which was declared a pandemic by the World Health Organization (WHO) in early 2020. This disease has caused a health and economic emergency worldwide. Currently, research on SARS-CoV-2 is booming and great efforts are being made to characterise SARS-CoV-2 molecularly. The genomic and molecular variability of SARS-CoV-2 can be the basis for glimpses of etiological and pathological aspects of this virus, understanding that the virus can accumulate mutations of importance while expanding worldwide, as well as also be able to establish antiviral strategies designed and based on the molecular specificities of SARS-CoV-2.

One of the most striking aspects of COVID-19 is the marked difference in the evolution of the disease in patients. The spread and manifestations of COVID-19, an infectious disease, are influenced by multiple interrelated factors. These include the virus itself (SARS-CoV-2), the human host (comorbidities and genetics), and the environment (physical conditions, social interactions, containment measures). All of these play a role in determining the course of the disease and the pandemic [1]. By elucidating and obtaining these genomic data, it would be possible to reveal the evolutionary events of SARS-CoV-2, establish the types of circulating genomes, and determine in which parts of the genome these viral isolates differ [2,3].

In Peru, lineages of regional and global relevance variants have emerged; some researchers detected the circulation of SARS-CoV-2 strains with the D614G mutation in the Lambayeque region at the beginning of 2020. This mutation had already spread widely in Europe at that time. However, other uncommon mutations demonstrate the virus’s rapid evolutionary processes and adaptive capabilities [4]. Subsequent investigations corroborated the presence of a variant endemic to the region, which was designated the Lambda variant [5].

The genetic variability of SARS-CoV-2 requires continuous study to elucidate various aspects of its molecular biology. Due to this, various modifications or changes in the nucleotide sequence of the viral genome have been reported worldwide, causing the appearance of variants, which have been grouped into differentiated clades. Among the variants of interest of SARS-CoV-2 are Lambda and Mu, which were first identified in Peru and Colombia, respectively. Meanwhile, the variants of concern of SARS-CoV-2 identified and reported globally, in chronological order, are Alpha (British), Beta (South African), Gamma (Brazilian), Delta (Indian) and recently Omicron [6].

For this reason, sequencing of the SARS-CoV-2 viral genome in Peru is urgently required; this will provide information on the prevalence of viral clades belonging to SARS-CoV-2, which could lead to a better understanding of transmission patterns, outbreak monitoring and formulation of effective containment measures. Mutation data may also provide important clues for developing vaccines, antiviral drugs, and effective diagnostic assays.

This study aimed to investigate the genomic variation in SARS-CoV-2 in Peru through the whole genome sequencing of SARS-CoV-2 strains and compare their evolutionary trajectories with global strains through phylogenetic analysis.

## 2. Methods

### 2.1. Sample Collection and Complete Sequencing of the SARS-CoV-2 Genome

Nasopharyngeal swabs were obtained from positive cases of COVID-19, and the first 30 sequenced samples were obtained at the Almanzor Aguinaga Asenjo Hospital belonging to Essalud, Chiclayo, Lambayeque, Peru, during the period from December 2020 to January 2021. The samples with a CT ≤28 were selected for genome sequencing; these samples (n = 30) were then sent to the Microbial Genomics laboratory of the Universidad Peruana Cayetano Heredia for subsequent analysis and sequencing. Another 44 samples of nasopharyngeal swabs from positive cases of COVID-19 confirmed by the reference laboratory of the Regional Health Management Lambayeque were included; these biological samples belong to the period from September to October 2021, thus completing 74 genomes sequenced by the authors of the present research.

Whole genome sequencing of SARS-CoV-2 isolates was performed with a MiSeq (Illumina, San Diego, CA, USA) using the COVIDseq Illumina kit. In detail, the samples were processed using the Sansure brand RNA kit and a Sansure Natch 48 automated device. Subsequently, genomic libraries were prepared with the Illumina COVIDSeq kit. The library sequencing was conducted using an instrument of the Illumina MiSeq model and the 300-cycle v2 kit to achieve an average coverage of 1500X. Once the sequences were obtained, they were processed and assembled with Illumina’s DRAGEN pipeline through its Base Space application. The Illumina MiSeq sequencing procedure was performed at the Microbial Genomics Laboratory, Department of Cellular and Molecular Sciences Faculty of Sciences and Philosophy (Universidad Peruana Cayetano Heredia, Lima, Peru).

### 2.2. Bioinformatics Analysis

All bioinformatic analyses used in the development of this research were based on the protocols proposed and designed for the genomic surveillance of SARS-CoV-2 in Costa Rica [7].

### 2.3. SARS-CoV-2 Sequences

Until 28 April 2022, all SARS-CoV-2 sequences available from the Lambayeque region of Peru were recovered and retrieved from the GISAID (Global Initiative on Sharing All Influenza Data, www.gisaid.org) database. A total of 714 sequences were recovered from the Lambayeque region, of which 74 were sequenced and uploaded to GISAID by the authors of this article.

### 2.4. Multi-Sequence Alignment

All sequences were aligned using MAFFT v7.471 [8]. The SARS-CoV-2 isolate Wuhan NC 045512.2 was used as a reference sequence to analyse mutations at the amino acid level.

### 2.5. Phylogenetic Analyses

The construction of the phylogenetic tree model was achieved with IQ-TREE v1.6.12 [9], including ModelFinder [10], to select the best nucleotide substitution model (using the Bayesian Information Criterion BIC, the best model was TN+F+I). The visualisation was performed using the iTOL v4 tool [11].

### 2.6. Selection and Identification of Mutations of Epidemiological Relevance of SARS-CoV-2 and Their Geographical Association

According to the table showing the top 5% mutations observed in the 714 genomes analysed in the Lambayeque region of Peru (Figure 1), a manual search was performed for the missense or non-synonymous mutations of each SARS-CoV-2 gene on the Nextstrain^®^ (https://nextstrain.org/) online server, identifying the lineage to which this genomic sequence belonged. Subsequently, this lineage was verified in Outbreak.info^®^ (https://outbreak.info/) to determine the geographical distribution of said mutations according to the GISAID database. Only those mutations predominant in Peru and mainly in the Lambayeque region were selected for detailed characterisation.

### 2.7. Ethical Considerations

Ethical approval for sample collection and analysis protocols was granted by the ethics committee of the Almanzor Aguinaga Asenjo Hospital, Chiclayo, Peru, through the ICIS-RPL code 066-DEC-2021. Participation in the study was voluntary, with the signing of an informed consent approved by the same ethics committee; in the case of patients, the consent was signed by their family member or proxy. All information obtained from participants was used only for this research. Therefore, such information will not be stored or used for further studies.

## 3. Results

Seventy-four nasopharyngeal swab samples were collected from SARS-CoV-2 positive patients (cycle threshold values [CT] obtained by qPCR, ≤28). The Illumina MiSeq sequencing procedure was performed in the Microbial Genomics Laboratory, Department of Cellular and Molecular Sciences Faculty of Sciences and Philosophy (Universidad Peruana Cayetano Heredia, Peru). The viral genomes were assembled by mapping the genome of the Wuhan Hu-1 strain deposited in GenBank as a reference (account number NC_045512).

Seventy-four genomic sequences were obtained from SARS-CoV-2 viral isolates from patients at the Almanzor Aguinaga Asenjo Hospital, Chiclayo, Peru. These sequences were divided into two batches of samples: the first 30 were obtained from COVID-19-positive nasopharyngeal swabs from December 2020 to January 2021. According to the PANGOLIN software v1, the 34 analysed sequences from the first batch were classified within the lineages C.14, C.33, B.1.1.485, B.1.1, B.1.1.1, and B.1.111. The C.14 lineage is the most predominant, at 76.7% (n = 23), and the other lineages range between 3.3–6.7%, respectively.

The second batch of samples consisted of 44 nasopharyngeal swabs positive for COVID-19 from September and October 2021. After analysing these sequences in Nextclade V1.10.0 (https://clades.nextstrain.org/) and PANGOLIN V3.1.16 (https://pangolin.cog-uk.io/), it was determined that the predominant variants in this sampling period were the Delta (72.7%), Gamma (13.6%), Mu (4.6%), and Lambda (9.1%) variants, which were distributed between clades 20J, 21G, 21H, 21J, and 21I.

Another variant of interest is C.37 (Lambda variant); the present investigation determined that 04 sequences (n = 4/44) of the second batch of samples analysed belonged to C.37, classified within clade 21G. C.37 is considered a variant native to Peru, also called the Andean variant; the first reports of C.37 began in Lima, Peru, in approximately August 2020. Subsequently, this variant has been predominant in the sequencing results detected in Peru since its first report, and it has spread to most countries in South America [12,13].

The sequences of the first and second batch of sequenced samples were analysed in the online programme Nextclade V1.10.0 (https://clades.nextstrain.org/) to assign the corresponding clades, identify the mutations in each SARS-CoV-2 gene, and also to perform quality control of these sequences.

Characteristic mutations that have been found in the C.14 lineage include the T1246I and G3278S mutations in the ORFIa gene; P314L in the ORFIb gene; D614G in spike protein; and R203K and G204R in gene N (Table 1).

In addition, in the C.14 lineage, it was possible to identify that the most frequent and relevant amino acid change in the SARS-CoV-2 Nucleocapsid gene was R203K and G204R (N gene, n = 34/35). Likewise, in the case of the changes in nearby nucleotides GGG>AAC at positions 28881–28883, the triplet was also present in n = 34/35 of the sequences analysed.

Using the Nextclade tool of Nextstrain, the Delta variant of SARS-CoV-2 (n = 32) was classified into clades 21J (n = 27) and 21I (n = 5). It was observed that within the Delta variant of SARS-CoV-2, there were several sublineages, among which were AY.26, AY.39.2, AY.100, AY.122, AY.43, AY.102, and B.1.617.2. The characteristic mutations found in the Delta variant and its sublineages are shown in Table 2.

As for the Gamma variant of SARS-CoV-2, according to the Nextclade tool of Nextstrain, the Gamma variant (n = 6) was classified in clade 20J. It was observed that within the Gamma variant, there were two sublineages, among which the following stand out: P.1 (n = 2) and P.1.12 (n = 4). The characteristic mutations found in the Gamma variant and its sublineages are shown in Table 3.

Concerning the Lambda variant of SARS-CoV-2, according to the Nextclade tool of Nextstrain, the Gamma variant (n = 4) was classified in clade 21G. The characteristic mutations found in the Lambda variant (C.37) are shown in Table 4.

A phylogenetic tree of the 714 genomes obtained on 28 April 2022 from the GISAID International Base (www.gisaid.org) of the Lambayeque region, Peru, was created. It can be seen in the phylogenetic tree that the sequences belong mainly to the variants Delta, Omicron, Mu, Lambda, and Gamma of SARS-CoV-2, as shown in Figure 2.

A potency law pattern was recognised in the analysed genomes, and the presence/absence of variants in the 714 sequences is evident. Few variants are widely distributed across genomes, and many are uniquely present in a single genome (Figure 1). This means that only a few variants are present in several genomes, and further analysis can focus on those variants.

The identification of variants of the SARS-CoV-2 genome observed in more than 5% of the genomes analysed in the Lambayeque region, Peru, was also carried out; a specific analysis of these variants is shown in Table 5. The identification of relevant mutations of SARS-CoV-2 and their geographical association was also carried out; for this, a manual review was conducted mutation by mutation in the Nextstrain online tool of those mutations with a frequency greater than 5% to determine where they had been reported according to the GISAID international database. A specific analysis was carried out if Peru emerged as one of the predominant countries for a particular mutation.

The K1817N mutations in the ORF1a gene, S74F in the ORF3a gene, and the P80R mutation in the N gene observed in the SARS-CoV-2 genomes from the Lambayeque region of Peru were selected for analysis because in Peru, a notable increase was found in these mutations during the period from July to August 2021, in contrast to the prevalence worldwide, which remains relatively constant and low.

For each of these mutations, its associated lineage or sublineage was determined; the prevalence statistics of that lineage or sublineage were also investigated worldwide, specifically in Peru. After carrying out all these searches, it was determined that the predominant SARS-CoV-2 sublineages in the 714 sequences analysed were AY.39.2 and P.1.12.1. It should be noted that both sublineages have been reported in various regions of Peru, with Lambayeque being the region with the highest prevalence (Figure 3 and Figure 4).

## 4. Discussion

Genomic surveillance of SARS-CoV-2 plays a critical role in understanding and responding to the pandemic. Tracking the emergence of mutations and variants through whole genome sequencing enables early detection of novel variants of concern and monitoring their spread, allowing public health officials to implement timely tailored containment measures. The study of the biological and pathogenic properties of new variants, including their transmissibility, virulence, and immune evasion ability, improves existing diagnostics and treatments to ensure they remain effective against new variants. Identifying key mutations correlating with concerning properties provides valuable insights into the virus’s adaptation. Therefore, genomic surveillance is essential to maintain pace with the evolution of SARS-CoV-2 and adapt strategies to combat current and future variants, thus improving our response to the pandemic [6].

The present study allowed us to identify and characterise the genome and lineage of 74 viral strains of the SARS-CoV-2 virus obtained from patients at the Almanzor Aguinaga Asenjo Hospital in Chiclayo, Peru, through next-generation sequencing (NGS) with the Illumina MiSeq system. NGS sequencing allows us to determine the molecular epidemiology of SARS-CoV-2 and learn about the virus’s evolution, transmission, virulence and pathology. Before the arrival of the COVID-19 pandemic, several researchers worldwide began to sequence the complete genome of SARS-CoV-2 to genetically understand this virus, try to elucidate its origin and find a molecular target that serves as a basis for the development of a biological product or vaccine against SARS-CoV-2. The implementation of bioinformatics and genomic tools allowed the active surveillance of SARS-CoV-2, as well as the identification of new lineages and the registration of new mutations in the viral genome, which will allow a better understanding of the evolution and replacement rates of the virus. Several countries in Latin America, Asia, Europe and Africa have published their sequencing results of the whole genome of SARS-CoV-2 in the GISAID international database [14,15].

The predominant lineage in this sampling period was lineage C.14 in 76.7% (n = 23) belonging to clade 20D, which also detected the circulation of lineages C.33, B.1.1.485, B.1.1, B.1.1.1, and B.1.111 in a percentage ranging between 3.3% and 6.7%, respectively, distributed between clades 20B and 20A. The results agreed with those reported by [4], who sequenced five genomes obtained from patients from the Lambayeque region at the end of April 2020, reporting the circulation of lineage B.1.1.1, which was classified according to Nextclade of Nextstrain in clade 20B. Also, our results agree with what was described by [3]. These authors indicate that SARS-CoV-2 isolates during the initial period of the pandemic in Peru belong or are grouped mainly in clade 20B; this clade is very characteristic of isolates obtained from patients with COVID-19 in the European continent. Likewise, these authors identified nine predominant lineages: A.1, A.2, A.5, B.1, B.1.1, B.1.1.1, B.1.5, B.1.8, and B.2, with the most predominant being B.1 and B.1.1.1.

According to the analyses carried out in GISAID and Pangolin [16] (B1.1.1, B1.5), the results highlight that most Peruvian SARS-CoV-2 sequences are classified within clade B.1 and within subclade B.1.1.1. The results described above differ from our results because the predominant lineage in our first sequences was C.14; this difference in results can be attributed to the sampling period in which the samples of nasopharyngeal swabs positive for COVID-19 were collected. Although there are no reports of the C.14 lineage, the GISAID-enabled outbreak.info mutation tracker (https://outbreak.info/situation-reports?pango=C.14) indicates that this lineage has been reported in the following countries: Peru (93.0%), United States (2.0%), Japan (2.0%), Democratic Republic of Congo (1.0%), and Brazil (1.0%) and was first reported on 20 March 2020.

The study reports that the principal variant detected in the second batch was B.1.617.2 (Delta), which has greater transmissibility virulence and can cause cases of reinfection and outbreaks due to the presence of a high number of mutations in the spike protein that allow more significant resistance to the action of antibodies or immune escape. The Delta variant has been reported in several countries worldwide and can replace other regional variants in circulation. The high infectivity of Delta is linked to its high viral load and the short incubation period until the appearance of symptoms. The study also reports that the Delta variant has been found to have immune evasion in patients who received doses of Pfizer^®^ (Pfizer Inc., New York, NY, USA), Moderna^®^ (Moderna, Inc. Cambridge, MA, USA), and Covax^®^, suggesting that the variant may require updated vaccines to provide better protection [17,18].

In addition, our sequences assigned or classified as Delta variant (21I, 21J) presented various mutations in the spike gene (S gene) such as L452R, T478K, D614G, and P618R. These mutations have been reported worldwide by various researchers, and indicate that they provide biological advantages, among which are: an increase in binding to the ACE-2 receptor, increased transmissibility, risk of hospitalisation, and immune escape or resistance to specific antibodies [19,20]. Some reports indicate that the Delta variant has undergone another mutation, K417N, T95I, and W258L, calling it the Delta Plus variant; however, in our results, we did not find this mutation in any of the analysed sequences belonging to the Delta variant. Some research indicates that the K417N, T95I, and W258L mutation of the spike protein increases the viral ability to achieve immune evasion; however, little is still known about the pathogenicity and virulence of this new variant of SARS-CoV-2 [21].

The study identified that the most frequent and relevant mutation in the SARS-CoV-2 spike gene of the C.14 lineage was D614G, which is associated with more significant pathogenesis and virulence, and evidence suggests that it can improve the transmission of the virus by increasing the amount of viral load in the upper respiratory tract. The sequences analysed and classified as C.37 contain a characteristic deletion in the gene S and mutations not synonymous in the gene spike. These could provide biological advantages such as increased transmissibility, virulence, viral invasion into host cells, and immune escape properties. The Gamma variant of SARS-CoV-2 was also detected in the study, which has lineage-defining mutations, including K417T, E484K, and N501Y, and mutations that allow this variant to increase ACE-2 receptor binding affinity, cause reinfection, increased transmissibility, higher viral load, and immune evasion [22]. The study also cites reports that patients infected with SARS-CoV-2 who carried the D614G mutation developed a moderate/severe COVID-19 condition, while patients infected with SARS-CoV-2 who did not carry this mutation developed mild symptoms and that the Gamma variant was the predominant lineage in the second wave of COVID-19 cases in Brazil.

During the third wave of the COVID-19 pandemic in Peru, the Delta and Gamma variants predominated until the emergence of the Omicron variant. Studies from other countries, such as one from Pakistan, report that the Delta, Beta, and Gamma variants had specific mutations that could provide various biological advantages to these variants. The simultaneous coexistence of highly transmissible SARS-CoV-2 variants could lead to evolutionary competition, where specific variants with mutations that improve their infectious capacity compete with others characterised by their immunological evasion capacity. In Peru, the Lambda variant (C.37) became the predominant variant in the coastal and Andean region, surpassing other circulating Variants of Concern (VOC) such as Gamma and Delta, despite Gamma having a higher frequency during the second wave in the Northwest region due to its proximity to Brazil (Vargas-Herrera et al., 2022a). The Lambda variant (identified as C.37) was initially identified in Peru in August 2020 where it became the predominant variant in early 2021. While it was detected to have a significant prevalence in Peru, its global spread was more limited than other variants, such as Delta and Omicron [23]. There are several possible explanations for Lambda’s initial prevalence in Peru and its lack of being widespread.

Local factors such as population density, mobility patterns, and social behaviours during restriction periods in Peru may have favoured Lambda variant transmission. It could also compete with other variants present in other regions with more extraordinary transmission and immune evasion capacity. Another factor was Peru’s effective implementation of public health measures that limited its expansion. Likewise, the Lambda variant probably needed the optimal characteristics to spread efficiently globally, given that its genetic properties and ability to infect cells and infect from person to person may have been less favourable than other variants. More research is required to fully understand the behaviour of this variant and why it spreads differentially in some regions.

The molecular mechanisms involved in the genetic variability of SARS-CoV-2 are multiple, and substitutions, insertions, deletions, and even genetic recombination mutations have been reported. It is known that the virus regions with the essential mutations reside in the spike protein (S) and the non-structural proteins (NSPs) such as ORF3b, ORF6, ORF7a, etc., which antagonise interferon signalling through different mechanisms and play a crucial role in altering the virus phenotype [24].

Some mutations, such as K417T, N501Y, and E484K, have been associated with higher affinity binding to the ACE2 receptor and resistance to neutralising antibodies, which could affect the infectivity and immune evasion of the virus. In addition, the error-prone replication process of the virus generates a broad spectrum of mutations, and dominant variants exhibit rapid propagation, contributing to the virus’s evolutionary dynamics and subsequent distribution [25]. The presence of specific mutations, such as Spike_D614G, has also been related to increased infectivity and transmissibility of the virus. These mutations can confer advantages by improving the ability of the virus to bind to host cells and evade immune responses, which could influence the appearance of mutations in virus variants [26].

It is known that the Beta (B.1.351) and Gamma (P.1) variants of SARS-CoV-2 present deletions in the spike protein region, such as the 242–244 deletion. These deletions can alter the virus’s interaction with cellular receptors and immune response. On the other hand, the Delta (B.1.617.2) variant shows the insertion of the L452R mutation in the spike protein, which seems to affect its interaction with ACE2 receptors and potentially increase its transmissibility. The Omicron (B.1.1.529) variant stands out for having many mutations, including insertions, deletions and multiple changes in the spike protein and other areas of the viral genome. An insertional genetic sequence in Omicron (insertion: ins214EPE) has even been detected that is not present in earlier versions of SARS-CoV-2. Still, it is common in other viruses, including those causing the common cold and in the human genome. By inserting this fragment into itself, Omicron could mimic the host and better evade the human immune system response, contributing to its transmissibility and the possibility of causing mild or asymptomatic disease [27].

In our results, we can also observe that the N gene of SARS-CoV-2 presents several amino acid mutations that confer various changes or biological advantages. Worldwide, several reports indicate that mutations in the N gene of SARS-CoV-2 reduce the sensitivity of molecular tests (RT-PCR) for the detection of SARS-CoV-2, thus causing the appearance of false negative results for this gene. The N gene of SARS-CoV-2 is of vital importance in the structure and viral cycle, as it is involved in viral assembly, replication and the immune response of the host; also, this SARS-CoV-2 gene is a gene not conserved due to its mutation rate. All these characteristics described above make this gene a target or target to update tests that allow viral diagnosis through and for the development of vaccines [28,29].

The natural evolution of SARS-CoV-2 has led to the emergence of multiple genetic variants with various biological properties, including increased transmission, immune escape, infectivity, and lethality. The initiation of mass vaccination could also be associated with an increase in selective pressure, leading to the appearance of escape mutants. Large-scale whole genome sequencing of SARS-CoV-2 is vital to track the spread of the virus, study local outbreaks, and identify critical mutations in SARS-CoV-2 genes. However, sharing sequencing results in the GISAID database is crucial for almost real-time genomic surveillance worldwide, providing a better understanding of the transmission and viral evolution dynamics of SARS-CoV-2 [30,31].

Finally, the importance of recognizing a power law pattern in the analysed genomes is fundamental as it allows us to identify critical mutations in the SARS-CoV-2 genes. This pattern shows the presence/absence of variants in the 714 genomes analysed. This pattern indicates that certain variants are widely distributed among the analysed genomes and many other variants are uniquely present in a single genome. This means that only a few variants are present in several genomes, and subsequent analysis can focus on those variants of epidemiological interest [7].

Although this study provides valuable information, it has some limitations related mainly to the sample size and the need for clinical data. Thus, only 74 viral sequences were analysed in two groups, which may limit the ability to detect some low-frequency circulating variants. The lack of clinical and epidemiological data associated with the cases analysed could be considered a limitation since the study focused on the genomic analysis of the samples but needed to report data on the severity of the cases, hospitalisations, contacts, etc. Although it was not the present study’s objective, this information is essential to determine the clinical and epidemiological impact of the detected variants. Lastly, it is necessary to consider a possible geographic bias since the study focused on a single city in the region; therefore, it may have captured only some of the diversity of variants circulating in other areas. Future genomic surveillance studies should ideally include representative samples from the entire region. Genomic surveillance in Peru, as in other countries of Latin America, has been vital in the understanding of the COVID-19 pandemic evolution during these almost four years [28,29,30].

Finally, regarding the Lambda variant, it is important to note that the global spread and dominance of specific SARS-CoV-2 variants can be influenced by various factors, including local epidemiological conditions, public health measures, vaccination rates, and the interplay between the virus and the host population. There are several potential reasons why the Lambda variant may have been predominant in Peru but not as widespread globally: (1) Local Factors: Regional differences in population density, healthcare infrastructure, and public health responses can impact the transmission and prevalence of a particular variant. Factors such as crowded living conditions, healthcare capacity, and adherence to preventive measures may contribute. (2) Competing Variants: other variants of concern, like Alpha, Beta, Delta, or others, may have outcompeted Lambda in other regions due to differences in transmissibility or immune escape properties during that time. (3) Vaccination Status: The spread of variants can also be influenced by the level of population immunity through vaccination. If other variants with a competitive advantage emerged in regions with higher vaccination coverage, they might have become more prevalent. (4) Human Mobility: International travel patterns and restrictions can impact the spread of variants. If a particular variant is prevalent in a country with limited international travel or strict entry protocols, its global dissemination may be limited. (5) Adaptation to Host Population: The success of a variant may also be influenced by its ability to adapt to the local population’s genetic and immune characteristics. It is essential to keep in mind that the situation with SARS-CoV-2 is dynamic, and ongoing research is continuously providing new insights. Monitoring the spread and impact of different variants is crucial for understanding the evolving nature of the virus and adapting public health strategies accordingly, including the importance of genomic surveillance and epidemiology [12,13,28,29,30], then continuing to make a molecular diagnosis of COVID-19 as a standard rule [31].

Genomic surveillance is a powerful tool for monitoring and understanding the dynamics of infectious diseases, enabling a more proactive and effective response to emerging threats, such as SARS-CoV-2 and future pandemic pathogens.

In conclusion, this study of SARS-CoV-2 genomes in Chiclayo, Peru, highlights the presence of multiple lineages and variants of concern circulating in the region. The emergence of Delta as the most common variant in later samples from 2021, along with other variants like Gamma, Mu, and Lambda, is particularly alarming due to their potential increased transmission and virulence and their possible ability to escape the immune response. The use of whole genome sequencing and data sharing in databases like GISAID is crucial for understanding the evolution and epidemiology of SARS-CoV-2, which can inform response measures and aid in detecting emerging strains. The findings underscore the importance of genomic surveillance in tracking the spread of SARS-CoV-2 variants and developing tailored public health strategies to limit their transmission. Continued monitoring and sequencing efforts are necessary to stay ahead of the virus’s evolution and ensure effective pandemic control.

## Figures and Tables

**Figure 1 tropicalmed-09-00046-f001:**
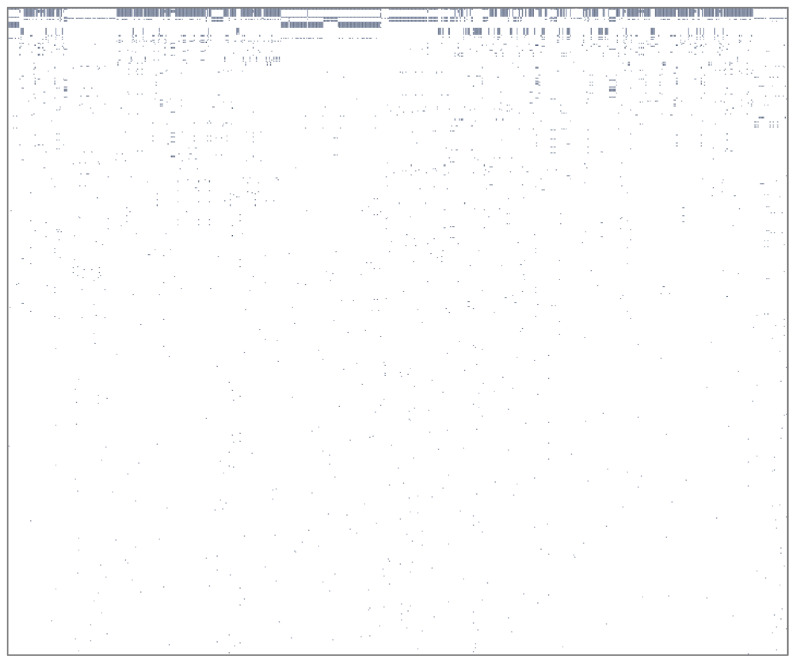
Law of potency of mutations observed in the analysis of 714 genomes from the Lambayeque region, Peru. The presence/absence of variants in the 714 genomes analysed is evident. Few variants are widely distributed among genomes, and many are uniquely present in a single genome.

**Figure 2 tropicalmed-09-00046-f002:**
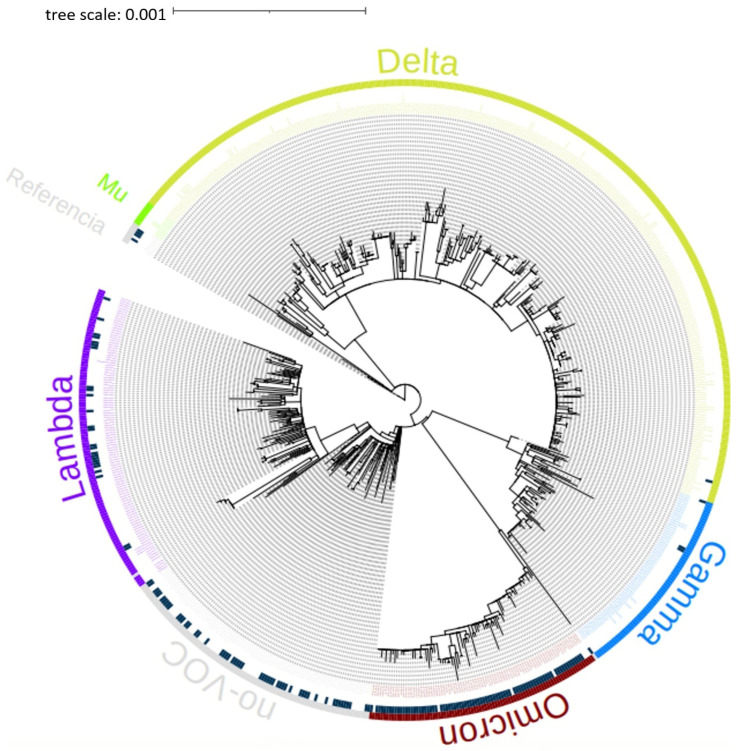
A phylogenetic tree was created in IQ-TREE v1.6.12 of the 714 genomes of the Lambayeque region, Peru (until 28 April 2022). The genomes are classified within the variants Mu, Delta, Gamma, Omicron, and Lambda.

**Figure 3 tropicalmed-09-00046-f003:**
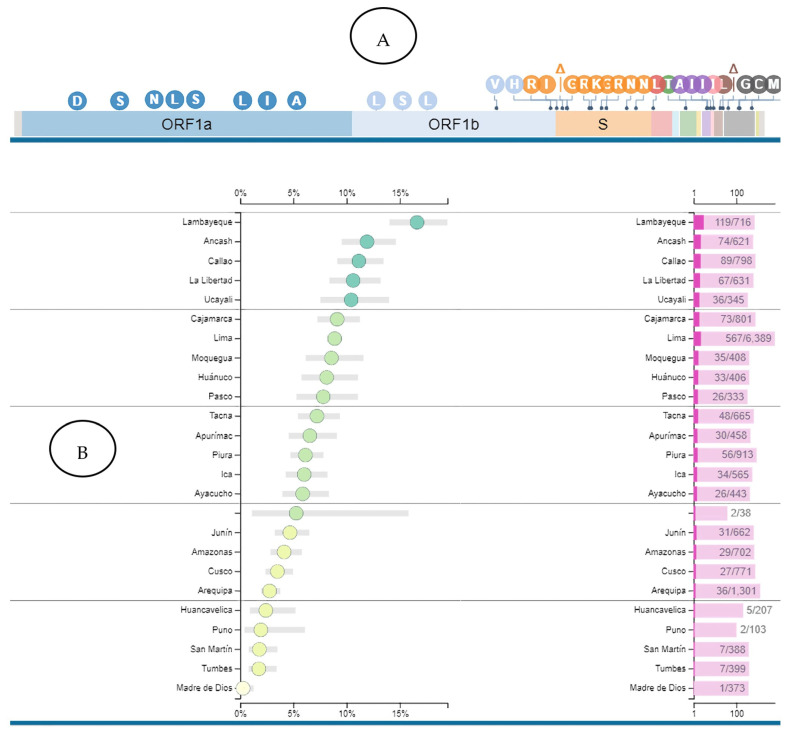
Epidemiological and genomic characteristics of the AY.39.2 lineage (**A**). The complete SARS-CoV-2 genome sequence, including the genes identified with different colours, is represented. The circles represent mutations of the AY.39.2 lineage. The lineage is characterised by various mutations distributed in the ORF1ab genes, S gene, ORF3a, and N gene. (**B**) The AY.39.2 lineage has been found in various regions of Peru, with Lambayeque being the region with the highest lineage prevalence. Image obtained from https://outbreak.info/situation-reports?xmin=2023-07-27&xmax=2024-01-27&pango=AY.39.2#geographic (accessed on 1 November 2023).

**Figure 4 tropicalmed-09-00046-f004:**
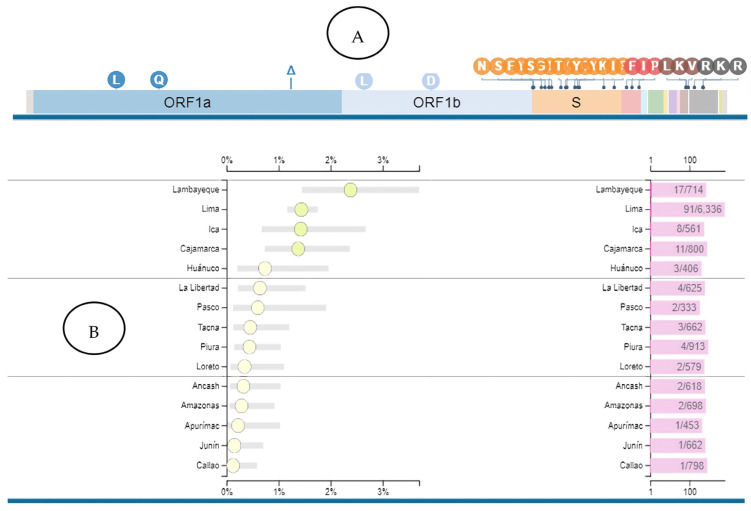
Epidemiological and genomic characteristics of the P.1.12.1 lineage (**A**). The complete SARS-CoV-2 genome sequence, including the genes identified with different colours, is represented. Table 1. 12.1 lineage. The lineage is characterised by various mutations distributed in the ORF1ab genes, S gene, ORF3a, and N gene. (**B**) The P.1.12.1 lineage has been found in various regions of Peru, with Lambayeque being the region with the highest lineage prevalence. Image obtained from https://outbreak.info/situation-reports?xmin=2023-07-27&xmax=2024-01-27&pango=P.1.12.1 (accessed on 1 November 2023).

**Table 1 tropicalmed-09-00046-t001:** Mutations found in the C.14 SARS-CoV-2 lineage of patients from the Lambayeque region, Peru.

Lineage C.14	Genes Affected by Mutations					
	ORF1a	ORF1b	S	ORF3a	ORF9b	N
C.14	P2144L T1246I G3278S P2685T	P314L S638I H1087Y V2073L	A222V D253E D614G	L101F L140F S171L V225F	T83I	H145Y R203K G204K

**Table 2 tropicalmed-09-00046-t002:** Mutations found in the sublineages of the Delta SARS-CoV-2 variant of patients from the Lambayeque region, Peru.

Gene	Sublineages of Delta Variant SARS-CoV-2						
	AY.26	AY.39.2	AY.122	AY.100	AY.43	AY.102	B.1.617.2
ORF1a	P1640L A3209V V3718A T3750I	E743D A1306S K1817N P2046L P2287S V2930L T3255I T3646A	K261N A1306S P2046L P2287S V2930L T3255I T3646A	T403I A1306S P2046L P2287S V2930L T3255I T3646A	A1306S P2046L P2287S V2930L T3255I T3646A	A1306S P2046L P2287S V2930L T3255I T3646A	A1306S T3255I T3646A
ORF1b	P314L G662S P1000L	P314L G662S P1000L A1918V Q2635H	P314L G662S P1000L A1918V	P314L G662S P1000L A1219S A1918V	P314L G662S L829I P1000L A1918V	P314L G662S P1000L A1918V	P314L G662S P1000L A1918V
S	T19R R158G Δ156/157 A222V L452R T478K D614G P681R D950N V1264L	T19R R158G Δ156/157 L452R T478K D614G P681R D950N K1073N	T19R R158G Δ156/157 L452R T478K D614G P681R D950N	T19R R158G Δ156/157 L452R T478K D614G P681R G769V D950N	T19R R158G Δ156/157 L452R T478K D614G P681R D950N	T19R R158G Δ156/157 L452R T478K D614G P681R D950N	T19R R158G Δ156/157 L452R T478K D614G P681R D950N
ORF3a	S26L	S26L	S26L	S26L	S26L T34A	S26L	S26L
M	I82T	I82T	I82T	I82T	I82T	I82T	I82T
ORF6	K48N	----	----	----	----	----	----
ORF7a	V82A T120I	V71I V82A T120I	V82A T120I	V82A T120I	V82A T120I	V82A T120I	V82A T120I
ORF7b	----	T40I	T40I	T40I	T40I	T40I	T40I
ORF 8	S84L Δ119/120	S84L Δ119/120	Δ119/120	Δ119/120	Δ119/120	Δ119/120	Δ119/120
N	D63G R203M D377Y	D63G R203M G215C D377Y	D63G R203M G215C D377Y	D63G R195K R203M G215C D377Y	Q9L D63G R203M G215C D377Y	D63G R203M G215C D377Y	D63G R203M G215C D377Y

**Table 3 tropicalmed-09-00046-t003:** Mutations found in the sublineages of the SARS-CoV-2 Gamma variant of patients from the Lambayeque region, Peru.

Gamma Variant Sublineages	Genes Affected by Mutations					
	ORF1a	ORF1b	S	ORF3a	ORF8	N
P.1.12	S1118L K1795Q Δ3675/3677	P314L E1264D	L18F T20N P26S D138Y R190S K417T N501Y D614G H655Y T1027I V1176F	S253P	E92K	P80R
P.1	S1118L K1795Q Δ3675/3677	P314L E1264D	L18F T20N P26S D138Y R190S K417T E484K N501Y D614G H655Y T1027I V1176F	S253P	E92K	P80R R203K G204R

**Table 4 tropicalmed-09-00046-t004:** Mutations found in the Lambda (C.37) SARS-CoV-2 variant from patients in the Lambayeque region, Peru.

Lambda Variant	Genes Affected by Mutations						
	ORF1a	ORF1b	S	ORF3a	ORF9b	M	N
C.37	T1246I P1659T P2287S F2387V P2483S L3201P T3255I G3278S A3620V Δ3675/3677	S59F P314L T1137I A1643V Y1784C K2385E K2674R	L5F G75V Δ246–252 L452Q A475V E484K P499R N501T D614G H655Y P681R T859N	P240H	P10S	I82T	P13L R203K G204R G214C

**Table 5 tropicalmed-09-00046-t005:** SARS-CoV-2 genome variants were observed in more than 5% of genomes analysed from the Lambayeque region, Peru.

Mutation Number	POS	REF	ALT	Total Number of Genomes with Mutation	Class of Mutation	Effect	Gene	Transcript	AA	Sequence in Transcript	Sequence Protein	Patterns in the World
53	3037	C	T	712	Synonymous	Low	ORF1ab	c.2772C>T	p.Phe924Phe	2772/21291	924/7096	N/A
490	23403	A	G	712	Missense	Moderate	S	c.1841A>G	p.Asp614Gly	1841/3822	614/1273	Disseminated in the world
236	10029	C	T	518	Missense	Moderate	ORF1ab	c.9764C>T	p.Thr3255Ile	9764/21291	3255/7096	Disseminated in the world
373	15451	G	A	330	Missense	Moderate	ORF1ab	c.15187G>A	p.Gly5063Ser	15187/21291	5063/7096	Disseminated in the world
541	25469	C	T	328	Missense	Moderate	ORF3a	c.77C>T	p.Ser26Leu	77/828	26/275	Disseminated in the world
644	28461	A	G	328	Missense	Moderate	N	c.188A>G	p.Asp63Gly	188/1260	63/419	Disseminated in the world
213	8986	C	T	309	Synonymous	Low	ORF1ab	c.8721C>T	p.Asp2907Asp	8721/21291	2907/7096	N/A
216	9053	G	T	309	Missense	Moderate	ORF1ab	c.8788G>T	p.Val2930Leu	8788/21291	2930/7096	Disseminated in the world
244	11332	A	G	309	Synonymous	Low	ORF1ab	c.11067A>G	p.Val3689Val	11067/21291	3689/7096	N/A
97	4181	G	T	308	Missense	Moderate	ORF1ab	c.3916G>T	p.Ala1306Ser	3916/21291	1306/7096	Disseminated in the world
624	28311	C	T	200	Missense	Moderate	N	c.38C>T	p.Pro13Leu	38/1260	13/419	Disseminated in the world
91	4002	C	T	194	Missense	Moderate	ORF1ab	c.3737C>T	p.Thr1246Ile	3737/21291	1246/7096	Very little disseminated in the world
157	5716	G	T	121	Missense	Moderate	ORF1ab	c.5451G>T	p.Lys1817Asn	5451/21291	1817/7096	Very little disseminated in the world
226	9867	T	C	115	Missense	Moderate	ORF1ab	c.9602T>C	p.Leu3201Pro	9602/21291	3201/7096	Very little disseminated in the world
225	9857	C	T	111	Synonymous	Low	ORF1ab	c.9592C>T	p.Leu3198Leu	9592/21291	3198/7096	N/A
508	25000	C	T	87	Synonymous	Low	S	c.3438C>T	p.Asp1146Asp	3438/3822	1146/1273	N/A
564	25584	C	T	87	Synonymous	Low	ORF3a	c.192C>T	p.Thr64Thr	192/828	64/275	N/A
137	5386	T	G	86	Synonymous	Low	ORF1ab	c.5121T>G	p.Ala1707Ala	5121/21291	1707/7096	N/A
259	11537	A	G	86	Missense	Moderate	ORF1ab	c.11272A>G	p.Ile3758Val	11272/21291	3758/7096	Disseminated in the world
338	13195	T	C	86	Synonymous	Low	ORF1ab	c.12930T>C	p.Val4310Val	12930/21291	4310/7096	N/A
604	26270	C	T	86	Missense	Moderate	E	c.26C>T	p.Thr9Ile	26/228	Set-75	Disseminated in the world
406	17259	G	T	72	Missense	Moderate	ORF1ab	c.16995G>T	p.Glu5665Asp	16995/21291	5665/7096	Very little disseminated in the world
153	5648	A	C	71	Missense	Moderate	ORF1ab	c.5383A>C	p.Lys1795Gln	5383/21291	1795/7096	Very little disseminated in the world
514	25088	G	T	71	Missense	Moderate	S	c.3526G>T	p.Val1176Phe	3526/3822	1176/1273	Very little disseminated in the world
14	733	T	C	70	Synonymous	Low	ORF1ab	c.468T>C	p.Asp156Asp	468/21291	156/7096	N/A
312	12778	C	T	70	Synonymous	Low	ORF1ab	c.12513C>T	p.Tyr4171Tyr	12513/21291	4171/7096	N/A
347	13860	C	T	70	Synonymous	Low	ORF1ab	c.13596C>T	p.Asp4532Asp	13596/21291	4532/7096	N/A
646	28512	C	G	70	Missense	Moderate	N	c.239C>G	p.Pro80Arg	239/1260	80/419	Very little disseminated in the world
31	1048	G	T	66	Missense	Moderate	ORF1ab	c.783G>T	p.Lys261Asn	783/21291	261/7096	Disseminated in the world
477	20937	G	T	58	Synonymous	Low	ORF1ab	c.20673G>T	p.Thr6891Thr	20673/21291	6891/7096	N/A
598	25844	C	T	44	Missense	Moderate	ORF3a	c.452C>T	p.Thr151Ile	452/828	151/275	Very little disseminated in the world
145	5515	G	T	41	Synonymous	Low	ORF1ab	c.5250G>T	p.Val1750Val	5250/21291	1750/7096	N/A
566	25613	C	T	38	Missense	Moderate	ORF3a	c.221C>T	p.Ser74Phe	221/828	74/275	Very little disseminated in the world

Disseminated in the world = high proportion of disseminated variants. Very little disseminated in the world = low proportion of disseminated variants.

## Data Availability

Available upon reasonable request.

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
