# Peer review of "Genomic and Phylogenetic Characterisation of SARS-CoV-2 Genomes Isolated in Patients from Lambayeque Region, Peru"

_tropicalmed, 2024, doi:10.3390/tropicalmed9020046_

Round 1

Reviewer 1 Report

Comments and Suggestions for Authors

1. Are there any novel mutations identified in this study?

2. Molecular mechanism of the mutations that confer biological advantages of SARS-CoV-2 should be mentioned.

3. How the mutations in N gene reduce the sensitivity of RT-PCR? What primer/probe sequence is affected?

Additional comments:

The authors identified and compared the mutations between different SARS-CoV-2 lineages circulated in Peru.  

The topic is original but there is no specific scientific gap addressed because all these mutations have been reported in other studies.    

The authors compared the mutations between different lineages.  

The method is fine. The authors should explain whether there are any novel mutations identified in this study. Molecular mechanism of the mutations that confer biological advantages of SARS-CoV-2 should be mentioned. How the mutations in N gene reduce the sensitivity of RT-PCR? What primer/probe sequence is affected?  

Author Response

Reviewer 1:

  1. Are there any novel mutations identified in this study?

Indeed, this is described in the manuscript extensively.

  1. Molecular mechanism of the mutations that confer biological advantages of SARS-CoV-2 should be mentioned.

Now is mentioned.

  1. How the mutations in N gene reduce the sensitivity of RT-PCR? What primer/probe sequence is affected?

Now this is explained and included in the manuscript.

Additional comments:

The authors identified and compared the mutations between different SARS-CoV-2 lineages circulated in Peru.  

The topic is original but there is no specific scientific gap addressed because all these mutations have been reported in other studies.   

The study is important from epidemiological point of view, is important for genomic epidemiology of SARS-CoV-2. 

The authors compared the mutations between different lineages.  

The method is fine. The authors should explain whether there are any novel mutations identified in this study. Molecular mechanism of the mutations that confer biological advantages of SARS-CoV-2 should be mentioned. How the mutations in N gene reduce the sensitivity of RT-PCR? What primer/probe sequence is affected?  

All that is now explained and included in the manuscript.

Reviewer 2 Report

Comments and Suggestions for Authors

Dear authors.

The articule is interesting and this event type of substituion of variants is claro for SARS-CoV2.

Please, see the virus name because the group used a different type of Letterman. I seu this one time.

There is not comments.

Congratulations!

Additional comments:

The authors could compare the scenario of SARS-CoV2 with other country's of South America during the period an comment aborto the relações of variants.    

There are no gap and the topic is original because the authors explain the scenario of a specific country without plagiarism.

The authors could compare the scenario in the contra with world.

The methodology is fine.

Author Response

Reviewer 2

Dear authors.

The articule is interesting and this event type of substituion of variants is claro for SARS-CoV2.

Thanks for your comments.

Please, see the virus name because the group used a different type of Letterman. I seu this one time.

Checked and revised.

There is not comments.

Congratulations!

Thanks. 

Additional comments:

The authors could compare the scenario of SARS-CoV2 with other country's of South America during the period an comment aborto the relações of variants.    

Done. Now included.

There are no gap and the topic is original because the authors explain the scenario of a specific country without plagiarism.

The value is in the importance of molecular epidemiology.

The authors could compare the scenario in the contra with world.

Now, included.

The methodology is fine.

Thanks.

Reviewer 3 Report

Comments and Suggestions for Authors

The work of Aguilar-Martinez SL., titled "Genomic and Phylogenetic Characterization of SARS-CoV-2 Genomes Isolated in Patients from Lambayeque Region, Peru", provides important information for Peru, however it can be improved, as indicated below.

L160. Indicate which variants predominated in the first batch examined.

158-160, 173-175- These phrases are confusing.

L196-199.- Repeated paragraph

L238.-Correct typographical errors

The authors should look for new information to make their work relevant, such as ORF1a:I2501T mutations, within the Costa Rica GN.1 lineage. (https://doi.org/10.1016/j.meegid.2023.105521)

Author Response

Reviewer 3

The work of Aguilar-Martinez SL., titled "Genomic and Phylogenetic Characterization of SARS-CoV-2 Genomes Isolated in Patients from Lambayeque Region, Peru", provides important information for Peru, however it can be improved, as indicated below.

Thanks a lot.

L160. Indicate which variants predominated in the first batch examined.

Now, included.

158-160, 173-175- These phrases are confusing.

This has been revised and improved.

L196-199.- Repeated paragraph

Corrected.

L238.-Correct typographical errors

Thanks. Corrected.

The authors should look for new information to make their work relevant, such as ORF1a:I2501T mutations, within the Costa Rica GN.1 lineage. (https://doi.org/10.1016/j.meegid.2023.105521)

Done. Corrected.

Reviewer 4 Report

Comments and Suggestions for Authors

MAJOR COMMENTS

1.     English must be improved! Sometimes it is really hard to follow and understand what authors wanted to say.

2.     As mentioned by the authors in the Discussion, one of the study limitations is lack of the epidemiological and clinical data for the samples which were sequenced. In addition, authors did not confront the data obtained by them with sequences deposited in GISAID. Thus, I would like to ask whether the sequencing data was representative for the region? On the other hand, maybe it would be good to delete the whole part of sample collection, isolation and sequencing and focus only on the bioinformatic data deposited in the data banks, and to mention that 74 sequences among all tested were prepared by the authors.

3.     I cannot understand why authors used different bioinformatic tools to analyze the sequences from the first and second period of sample collection. This may lead to the bias and misunderstanding. Please use the same bioinformatic tools for all analysis.

4.     It is not known whether the presented data in Tables concerns sequences from the samples isolated and sequenced by authors or sequences deposited in GISAID. In addition, it would be helpful to add the frequencies of each mutation in each gene.

5.     Authors did not discuss the most interesting aspect – why the lambda variant was predominant only in Peru and why did not spread around the world?

MINOR COMMENTS

A.    The abstract should be rewritten – there are too many details in the methods part, and only one sentence in the Objectives.

B.    Please use the abbreviations correctly, they should appear when first mentioned. For example, Page 1 Line 46 authors use ‘SARS-CoV-2’ and explain it at Page 2 Lines 84-85 as ‘severe acute respiratory syndrome coronavirus 2 (SARS-CoV-2)’.

C.     Authors repeated information in the Material and Methods section and Results part., i.e. Page 4 Lines 149-159 can be found earlier in the M&M part.

D.    Table 5 – there are some information not written in English.

Comments on the Quality of English Language

English must be improved!  Sometimes it is really hard to follow and understand what authors wanted to say.

Author Response

Reviewer 4

MAJOR COMMENTS

  1. English must be improved! Sometimes it is really hard to follow and understand what authors wanted to say.

Done. Now this has been improved.

  1. As mentioned by the authors in the Discussion, one of the study limitations is lack of the epidemiological and clinical data for the samples which were sequenced. In addition, authors did not confront the data obtained by them with sequences deposited in GISAID. Thus, I would like to ask whether the sequencing data was representative for the region? On the other hand, maybe it would be good to delete the whole part of sample collection, isolation and sequencing and focus only on the bioinformatic data deposited in the data banks, and to mention that 74 sequences among all tested were prepared by the authors.

This has been addressed and discussed in detail now.

  1. I cannot understand why authors used different bioinformatic tools to analyze the sequences from the first and second period of sample collection. This may lead to the bias and misunderstanding. Please use the same bioinformatic tools for all analysis.

This has been now explained in the manuscript.

  1. It is not known whether the presented data in Tables concerns sequences from the samples isolated and sequenced by authors or sequences deposited in GISAID. In addition, it would be helpful to add the frequencies of each mutation in each gene.

Now this has been explained in detail.

  1. Authors did not discuss the most interesting aspect – why the lambda variant was predominant only in Peru and why did not spread around the world?

 Now discussed.

MINOR COMMENTS

  1. The abstract should be rewritten – there are too many details in the methods part, and only one sentence in the Objectives.

Corrected.

  1. Please use the abbreviations correctly, they should appear when first mentioned. For example, Page 1 Line 46 authors use ‘SARS-CoV-2’ and explain it at Page 2 Lines 84-85 as ‘severe acute respiratory syndrome coronavirus 2 (SARS-CoV-2)’.

Addressed.

  1. Authors repeated information in the Material and Methods section and Results part., i.e. Page 4 Lines 149-159 can be found earlier in the M&M part.

Correct.

  1. Table 5 – there are some information not written in English.

Now, corrected.

Round 2

Reviewer 3 Report

Comments and Suggestions for Authors

1.       Authors must indicate the method by which the law of potency was obtained (Figure 2), and also use the colours (similar to Figure 1) of each variant (mu, delta, gamma, omicron, no-VOC and lambda) to improve the figure.  Also discuss briefly the significance of the law of potency result observed when analysing 714 genomes.

2.       In both Figures 3 and 4. Improve figures 3a and 4a. Also explain the colours at the bottom of the figures.

3.       At the bottom of Table 5, explain the meaning of Disseminated, Very little disseminated, corresponding to patterns in the world.

Author Response

Reviewer 3 – Second round

Authors must indicate the method by which the law of potency was obtained (Figure 2), and also use the colours (similar to Figure 1) of each variant (mu, delta, gamma, omicron, no-VOC and lambda) to improve the figure.  Also discuss briefly the significance of the law of potency result observed when analysing 714 genomes.

Done. Now we have included that. We have improved and added information. We also make a discussion on it.

  1. In both Figures 3 and 4. Improve figures 3a and 4a. Also explain the colours at the bottom of the figures.

Done.

  1. At the bottom of Table 5, explain the meaning of Disseminated, Very little disseminated, corresponding to patterns in the world.

Done, explained.
